# An Eco-Driving Strategy at Multiple Fixed-Time Signalized Intersections Considering Traffic Flow Effects

**DOI:** 10.3390/s24196356

**Published:** 2024-09-30

**Authors:** Huinian Wang, Junbin Guo, Jingyao Wang, Jinghua Guo

**Affiliations:** 1Department of Mechanical and Electrical Engineering, Xiamen University, Xiamen 361005, China; 19920220156448@stu.xmu.edu.cn; 2Department of Missile Engineering, Rocket Force University of Engineering, Xi’an 710025, China; gib_202@163.com; 3Department of Automation, Xiamen University, Xiamen 361005, China; wangjingyao1@xmu.edu.cn

**Keywords:** eco-driving, traffic flow, multiple signalized intersections, genetic algorithm, connected vehicle

## Abstract

To encourage energy saving and emission reduction and improve traffic efficiency in the multiple signalized intersections area, an eco-driving strategy for connected and automated vehicles (CAVs) considering the effects of traffic flow is proposed for the mixed traffic environment. Firstly, the formation and dissipation process of signalized intersection queues are analyzed based on traffic wave theory, and a traffic flow situation estimation model is constructed, which can estimate intersection queue length and rear obstructed fleet length. Secondly, a feasible speed set calculation method for multiple signalized intersections is proposed to enable vehicles to pass through intersections without stopping and obstructing the following vehicles, adopting a trigonometric profile to generate smooth speed trajectory to ensure good riding comfort, and the speed trajectory is optimized with comprehensive consideration of fuel consumption, emissions, and traffic efficiency costs. Finally, the effectiveness of the strategy is verified. The results show that traffic performance and fuel consumption benefits increase as the penetration rate of CAVs increases. When all vehicles on the road are CAVs, the proposed strategy can increase the average speed by 9.5%, reduce the number of stops by 78.2%, reduce the stopped delay by 82.0%, and reduce the fuel consumption, NO_x_, and HC emissions by 20.4%, 39.4%, and 46.6%, respectively.

## 1. Introduction

The energy usage and pollutants produced by transportation have a significant effect on the environment, hence the sustainability of transportation systems has become a primary strategic objective for many countries [1,2]. The frequent idling and start-stop of vehicles due to the presence of intersection signals leads to frequent traffic congestion at intersections, resulting in intersections becoming a crucial area for energy usage and pollutant emissions in the road network [3,4,5]. With the development of CAV technology, CAVs can establish connections with roadside infrastructure and other vehicles to receive information about the operating status of vehicles on the road and SPaT (Signal Phase and Timing) information to assist vehicles in making decisions to cross signalized intersections safely, efficiently, and energy-efficiently [6,7,8,9]. The eco-driving strategy has enormous potential to improve traffic efficiency and reduce energy consumption and emissions [10,11]. Meanwhile, urban road intersections are usually in the form of intersection clusters, so the research on eco-driving strategies through multiple intersections is significant.

Therefore, this paper proposes an eco-driving speed trajectory optimization strategy considering the effects of traffic flow. The main contributions are as follows.

(I)Considering the effects of queuing on CAVs and the obstruction of CAVs to following vehicles in a mixed environment, a traffic flow situation estimation model is constructed that can estimate multiple signalized intersection queue lengths and obstructed fleet lengths.(II)A feasible speed set calculation method for multiple signalized intersections based on traffic flow situation estimation is proposed, which can calculate the speed range that enables vehicles to cross the multiple signalized intersections without stopping and obstructing the following vehicles.(III)Optimizing CAV speed trajectories at multiple signalized intersections based on a genetic algorithm, aiming to find the optimal speed trajectory in the feasible speed set that maximizes traffic efficiency while minimizing fuel consumption and emissions.

The rest of this paper is as follows: in Section 2, an overview of the eco-driving strategy is discussed; Section 3 displays the traffic flow situation estimation model; the proposed strategy is constructed in Section 4; the effectiveness of the strategy is experimentally verified in Section 5; finally, Section 6 contains the conclusions.

## 2. Related Work

The eco-driving strategy has been researched by scholars. A signalized intersection speed planning method by designing a two-layer framework that first determines a target speed profile scenario based on vehicle speed, position, and SPaT information is proposed in [12], then treats the problem as a speed planning problem with speed and acceleration limitations and solves it to produce a speed trajectory with a minimal jerk; however, only isolated intersection problems were considered. Ref. [13] develops a partially automated vehicle system for eco-crossing signalized intersections that receive SPaT information and automatically follows recommended speed profiles. Experiments in real vehicles showed that automatic following can reduce fuel consumption compared to manual following, but did not consider coupling issues in multiple intersections. An isolated signalized intersection eco-driving system that prioritizes traffic efficiency is proposed in [14], which solves the optimal control problem and verifies the robustness of the system under different connected vehicles’ penetration rates. However, fuel consumption and emissions were not considered in the strategy. Ref. [15] considers powertrain dynamics to study the fuel-optimal multi-intersection eco-driving operating rules for connected vehicles and proposes an optimal control problem solution approach using the Legendre Pseudospectral (LPS) algorithm to minimize engine fuel consumption, but emissions were not considered. Ref. [16] considers the uncertainty of SPaT and approaches the eco-driving problem as a chance-constrained optimization problem. The dynamic programming approach is employed to solve it, along with a data-driven approach, which could potentially lead to interpretability issues. The aforementioned literature uses information on connected vehicle status and SPaT to optimize the vehicle speed trajectory but ignores the interaction between vehicles, which may lead to a large gap between the actual effect of the strategy and the theoretical analysis results.

The Eco-Cooperative Adaptive Cruise Control (Eco-CACC) method for CAVs passing through a multiple intersection with improved dynamic planning algorithms is proposed in [17], which reduces energy consumption by about 8% compared to manual driving; however, the cost function only considered energy consumption and did not account for traffic efficiency and emissions. A cooperative eco-driving control model in a mixed traffic environment is proposed in [18], focusing on the influence of the penetration rate, classifying the vehicles into leader and follower; however, the paper only considered enabling vehicles to pass through intersections without stopping, without optimizing speed based on energy consumption and emissions. A decentralized fuel economic control approach used to increase the mobility and energy utilization of a fleet, assuming that the vehicles have the availability of SPaT information and the statuses of neighboring vehicles, is proposed in [19], and the fast-MPC strategy is used to speed up the computation, but cannot guarantee comfort. An intelligent driver model that combines SPaT information with an adaptive cruise control (ACC) system is proposed in [20], which treats red lights as dummy preceding cars, considers queuing behavior before intersections, but does not consider further improving traffic efficiency and optimizing speed trajectory for energy consumption and emissions reduction. An eco-driving algorithm that considers vehicle queue information as well as SPaT information is proposed in [21]. It analyzed the algorithm’s benefits under different market penetration rates, demand levels, and phase splits but did not consider the obstruction of following vehicles by connected vehicles. Ref. [22] proposes hybrid reinforcement learning eco-driving algorithms to design deep deterministic policy gradient algorithms to learn the speed trajectory of vehicles through intersections to decrease fuel consumption while ensuring traffic efficiency and to develop deep Q-learning algorithms for lane-changing decisions. Ref. [23] proposed a reinforcement learning (RL)-based CAV car-following model for signalized intersections. However, reinforcement learning-based approaches often require detailed environment models, heavily rely on configuration and parameter tuning, involve extensive interaction and training time, and have limited interpretability. Ref. [24] develops an eco-driving strategy for urban scenarios using reinforcement learning, considering the non-deterministic behavior of other vehicles as part of the strategy but not explicitly considering the impact on following vehicles. Considering the interactions between vehicles helps improve the signalized intersection scenario portrayal. However, the existing research only considers the effects of queuing on the connected vehicles, and there is less research on the obstruction of the CAVs to the following vehicles. Most existing research focuses on the optimization objectives of traffic efficiency and energy usage without considering the impact of emissions.

## 3. Estimation of Traffic Flow Situation at Signalized Intersection

Since the formation and dissipation process of the queue in front of the intersection will affect the trajectory of the CAVs crossing the intersection, the planning of vehicle speed trajectories through the intersection is required to consider the SPaT information and the queuing of vehicles at intersections. At the same time, CAVs may obstruct following vehicles crossing the intersection at the same phase; therefore, the speed trajectory planning is required to consider not only their own efficiency and energy consumption but also the effects on the following vehicles. In this paper, the queue length, the queue dissipation time, and the length of the rear obstructed fleet are estimated based on the traffic flow density, and then the vehicles are planned with a reasonable speed trajectory based on the estimated information so that the queue has dissipated when the vehicles cross the intersection and does not obstruct the following vehicles.

The Lighthill–Whitham–Richard (LWR) theory [25,26] gives the basic relationship between traffic wave flow and concentration, reflecting the relationship between traffic flow characteristic variables, as shown in Figure 1. The traffic concentration is *ρ*_0_, the flow is *q*_0_ upstream of the intersection, and the traffic state is at point A. When the signal light is red, a queue will be formed in front of the intersection, as shown in Figure 2; at this time, the traffic concentration reaches a maximum value *ρ_j_*, the traffic flow is 0, and the traffic state is at point C in Figure 1. When the signal light turns green, the queue will dissipate with the saturated flow, and at this time, the traffic state downstream of the intersection is at point B in Figure 1, the traffic concentration is *ρ*_c_, and the flow is *q*_c_.

The queue formation process in front of the signalized intersection is the process of transferring from the state of point *A* to the state of point *C*. Based on the LWR theory, the shock wave speed during queue formation is computed as
(1)$$v_{AC}=\frac{q_{0}}{\rho_{0}-\rho_{j}}$$

The queue dissipation process is the transfer process from state *C* to state *B*. The rarefaction wave speed of this process is computed as
(2)$$v_{CB}=\frac{q_{c}}{\rho_{c}-\rho_{j}}$$

The maximum queue length in front of the intersection estimated as
(3)$$l_{q\max}=\frac{t_{g}-t_{r}}{v_{AC}-v_{CB}}\times v_{CB}\times v_{AC}$$
where *t_g_* is the instant of the signal light turning green, *t_r_* is the start instant of the signal light turning red.

Assuming that the vehicle arrives at the intersection at instant *t*_0_ and travels at speed *v*_0_, the queue length estimated as
(4)$$l_{q}=\left\{\begin{array}{cc} \frac{v_{AC}}{v_{AC}-v_{0}}\times \left[l-v_{0}(t_{r}-t_{0})\right] & t_{0}\in \left[t_{r}-\frac{l}{v_{0}},t_{g}+\frac{t_{g}-t_{r}}{v_{CB}-v_{AC}}\times v_{AC}-\frac{l-l_{q\max}}{v_{0}}\right]\\ 0 & t_{0}\notin \left[t_{r}-\frac{l}{v_{0}},t_{g}+\frac{t_{g}-t_{r}}{v_{CB}-v_{AC}}\times v_{AC}-\frac{l-l_{q\max}}{v_{0}}\right] \end{array}\right.$$
where *l* is the distance from CAV to the downstream intersection at instant *t*_0_.

The queue dissipation time can be estimated as
(5)$$t_{q}=\frac{l_{q}}{v_{CB}}$$

As shown in Figure 3, when a vehicle crosses through an intersection, it may force following vehicles to fail to cross the intersection at the current phase, so it is essential to reserve some space for following vehicles. The following vehicles that can cross the intersection during the present phase through speed optimization are defined as possible obstructed vehicles. Assuming that the possible obstructed vehicles travel at the maximum speed vcmax, the length of the obstructed fleet is estimated as
(6)$$l_{h}=\left\{\begin{array}{cc} (t_{r}+\left(x-1\right)\times T-\frac{l}{v_{c\max}}-t_{0})\times q_{0}\times d_{n} & t_{0}\leq t_{r}+\left(x-1\right)\times T-\frac{l}{v_{c\max}}\\ 0 & t_{0}\geq t_{r}+\left(x-1\right)\times T-\frac{l}{v_{c\max}} \end{array}\right.$$
where *T* is the duration of one signal cycle, *d_n_* is the sum of vehicle length and the minimum following distance, which can be calculated based on the following model.

## 4. Eco-Driving Strategy Considering Traffic Flow Effects

### 4.1. System Architecture

In this paper, the traffic environment is mixed. As shown in Figure 4, the roadside node can transmit real-time traffic flow information and SPaT information detected by the roadside facilities to the cloud server. The cloud server identifies, tracks, and estimates traffic flow situation and traffic participating object status in real-time based on the interaction information of communication nodes and connected vehicles for use in CAV decision-making. The speed trajectory obtained by connected vehicles through decision-making is executed by the autonomous driving system. Therefore, when CAV approaches the intersection area, the eco-driving strategy needs to determine the speed trajectory based on the relative position of CAV to downstream signalized intersections, traffic flow information, the speed of CAV, and SPaT information.

The following hypotheses were established for this study.

No overtaking or lane changing.The CAVs completely follow the speed trajectory of the proposed strategy under safe conditions.The effect of the slope on the vehicle is ignored.

### 4.2. Feasible Speed Set for Multiple Signalized Intersection

When the CAV enters the intersection, it determines whether the vehicle can pass the downstream intersection without stopping through speed trajectory optimization based on various information.

Assuming that *l_k_* is the distance from CAV to the *k*th downstream intersection at instant *t*_0_, k=1,2,3,⋯,∞, *t*_g*kx*_ is the start instant of the green light of the *x*th signal cycle at the *k*th intersection, x=1,2,3,⋯,∞, and *t_rkx_* is the start instant of the red light. That is, the *t*_g*kx*_~*t_rkx_* in the *x*th signal cycle of the *k*th intersection is a green phase, as shown in Figure 5. When there are no other vehicles on the road, the set of speeds that would allow a vehicle to cross the *k*th intersection without stopping is
(7)$$\left[\frac{l_{k}}{t_{rk1}-t_{0}},\frac{l_{k}}{t_{gk1}-t_{0}}\right]\cup \left[\frac{l_{k}}{t_{rk2}-t_{0}},\frac{l_{k}}{t_{gk2}-t_{0}}\right]\cup \cdots \cup \left[\frac{l_{k}}{t_{rkx}-t_{0}},\frac{l_{k}}{t_{gkx}-t_{0}}\right]$$

The speed set is the average speed of the CAV from its current location to the *k*th intersection. They will set up a speed profile later.

Assuming that the minimum planning speed is *v*_min_ and the maximum speed limit is *v*_max_, considering the queuing of vehicles in front of the intersection and the obstruction to the following vehicle, the feasible speed set for a vehicle to pass the *k*th intersection without stopping and without obstructing the following vehicles is
(8)$$V_{k}=\left[v_{\min},v_{\max}\right]\cap \left[\frac{l_{k}+l_{hk1}}{t_{rk1}-t_{0}},\frac{l_{k}-l_{qk1}}{t_{gk1}+t_{qk1}-t_{0}}\right]\cup \left[\frac{l_{k}+l_{hk2}}{t_{rk2}-t_{0}},\frac{l_{k}-l_{qk2}}{t_{gk2}+t_{qk2}-t_{0}}\right]\cup \cdots \cup \left[\frac{l_{k}+l_{hkx}}{t_{rkx}-t_{0}},\frac{l_{k}-l_{qkx}}{t_{gkx}+t_{qkx}-t_{0}}\right]$$
where *l_hkx_* is the length of the obstructed fleet, which can be calculated by using Equation (6); *l_qkx_* is the queue length, which can be calculated by using Equation (4); and *t_qkx_* is the queue dissipation time when the vehicle reaches the *k*th intersection in the *x*th signal cycle, which can be calculated by using Equation (5).

When *V*_1_ is not empty, it means that the vehicle can cross the first downstream intersection without stopping and without obstructing the following vehicles by speed optimization. At this point, calculate the feasible speed set *V*_2_, and determine whether the set is empty. When there is $V_{1}\cap V_{2}\cap \cdots \cap V_{k}\neq \emptyset$ and $V_{1}\cap V_{2}\cap \cdots \cap V_{k}\cap V_{k+1}=\emptyset$ indicates that the vehicle can only cross the 1st to the kth downstream intersection without stopping and without obstructing the following vehicles through one speed adjustment, and the feasible speed set through the 1st to the *k*th intersection is $V_{1}\cap V_{2}\cap \cdots \cap V_{k}$. When the vehicle has passed the kth intersection, CAVs should be planned again for the subsequent intersection operation strategy. When *V*_1_ is an empty set, the vehicle approaches the intersection at its current speed and stops to wait to cross it.

### 4.3. Velocity Trajectory Based on Trigonometric Profile

To take the fuel saving and emission reduction and the comfort demand of the passengers into account, this paper uses the trigonometric profile to realize the speed variation with the following equation
(9)$$\left\{\begin{array}{cc} v_{c}-(v_{c}-v_{0})\cos(\alpha t) & 0\leq t\leq \frac{\pi }{2\alpha }\\ v_{c}-(v_{c}-v_{0})\frac{\alpha }{\beta }\cos\beta (t-\frac{\pi }{2\alpha }+\frac{\pi }{2\beta }) & \frac{\pi }{2\alpha }\leq t\leq \frac{\pi }{2\alpha }+\frac{\pi }{2\beta }\\ v_{c}+(v_{c}-v_{0})\frac{\alpha }{\beta } & \frac{\pi }{2\alpha }+\frac{\pi }{2\beta }\leq t\leq \frac{L}{v_{c}} \end{array}\right.$$
where *v*_0_ is the original speed, *v_c_* is the target speed, *α* and *β* are the parameters of the speed trajectory, *α* is related to the change rate of acceleration in the M region and *β* is related to the change rate of acceleration in the N region, as shown in Figure 6, *L* is the speed planning distance, when the vehicle is affected by the queue, the planning distance is the distance from CAV to the end of the queue, when the vehicle obstructs the following vehicle, the planning distance is the distance between CAV and the intersection plus the distance reserved for the obstructed vehicles, otherwise the planning distance is the distance between CAV and the intersection, the calculation formula is as follows.
(10)$$L=\left\{\begin{array}{cc} l_{k}-l_{qkx} & \frac{\pi }{2\alpha_{0}}+\frac{\pi }{2\beta_{0}}+\frac{l_{k}-l_{qkx}-\int_{0}^{\frac{\pi }{2\alpha_{0}}}\left(v_{c}-(v_{c}-v_{0})\cos(\alpha_{0}t)\right)dt}{v_{c}+(v_{c}-v_{0})\frac{\alpha_{0}}{\beta_{0}}}-\frac{\int_{\frac{\pi }{2\alpha_{0}}}^{\frac{\pi }{2\alpha_{0}}+\frac{\pi }{2\beta_{0}}}\left(v_{c}-(v_{c}-v_{0})\frac{\alpha_{0}}{\beta_{0}}\cos\beta_{0}(t-\frac{\pi }{2\alpha_{0}}+\frac{\pi }{2\beta_{0}})\right)dt}{v_{c}+(v_{c}-v_{0})\frac{\alpha_{0}}{\beta_{0}}}\leq t_{gkx}+t_{qkx}-t_{0}\\ l_{k}+l_{hkx} & \frac{\pi }{2\alpha_{0}}+\frac{\pi }{2\beta_{0}}+\frac{l_{k}+l_{hkx}-\int_{0}^{\frac{\pi }{2\alpha_{0}}}\left(v_{c}-(v_{c}-v_{0})\cos(\alpha_{0}t)\right)dt}{v_{c}+(v_{c}-v_{0})\frac{\alpha_{0}}{\beta_{0}}}-\frac{\int_{\frac{\pi }{2\alpha_{0}}}^{\frac{\pi }{2\alpha_{0}}+\frac{\pi }{2\beta_{0}}}\left(v_{c}-(v_{c}-v_{0})\frac{\alpha_{0}}{\beta_{0}}\cos\beta_{0}(t-\frac{\pi }{2\alpha_{0}}+\frac{\pi }{2\beta_{0}})\right)dt}{v_{c}+(v_{c}-v_{0})\frac{\alpha_{0}}{\beta_{0}}}\geq t_{rkx}-t_{0}\\ l_{k} & other \end{array}\right.$$
where *α*_0_ and *β*_0_ are calculated when the planning distance is the distance between CAV and the intersection.

To guarantee that the vehicle arrives at the planned location at the specified time, it is necessary to ensure that the integration of speed over time is equal to the planning distance *L*, combined with the vehicle performance limitations and comfort requirements, and a set of constraints equations is established.
(11)$$\left\{\begin{array}{c} L=\int_{0}^{\frac{\pi }{2\alpha }}\left(v_{c}-(v_{c}-v_{0})\cos(\alpha t)\right)dt+\int_{\frac{\pi }{2\alpha }}^{\frac{\pi }{2\alpha }+\frac{\pi }{2\beta }}\left(v_{c}-(v_{c}-v_{0})\frac{\alpha }{\beta }\cos\beta (t-\frac{\pi }{2\alpha }+\frac{\pi }{2\beta })\right)dt+(v_{c}+(v_{c}-v_{0})\frac{\alpha }{\beta })(\frac{L}{v_{c}}-\frac{\pi }{2\alpha }-\frac{\pi }{2\beta })\\ \left|J\right|=\left|(v_{c}-v_{0})\alpha \beta \right|\leq 10\\ \left|a_{\max}\right|\leq 2.5\\ v_{\min}\leq v\leq v_{\max}\\ \alpha =\max\left\{\alpha \right\} \end{array}\right.$$
where *J* is the change rate of acceleration, *a*_max_ is the maximum acceleration, and the maximum acceleration considering the comfort demand of the passengers should not exceed 2.5 m∙s^−2^, while the absolute value of *J* should not exceed 10 m∙s^−3^. To make the vehicle reach a constant speed as soon as possible, α takes the maximum feasible value. Therefore, according to the formula and constraints, the speed trajectory can be uniquely determined when the *v*_0_ and the *v_c_* are determined.

### 4.4. Fuel Consumption and Emission Calculation

The control method in this paper is to control the speed trajectory of the CAVs one by one, and the VT-Micro model based on speed acceleration is selected to estimate emissions and fuel consumption of vehicles crossing through the speed planning section with different speed trajectories. The VT-Micro model uses the concept of Measure of Effectiveness (*MOE*) when calculating fuel consumption and emissions. The calculation formula for *MOE* is:(12)$$MOE_{e}=\exp\left(\sum_{i=0}^{3}\sum_{j=0}^{3}k_{i,j}^{e}v^{i}a^{j}\right)$$
where *MOE*_e_ is the vehicle fuel consumption rate or emission rate; *i* is the power index of speed; *j* is the power index of acceleration; $k_{i,j}^{\left(e\right)}$ is the regression coefficient [27,28].

As shown in Figure 7, CAVs can cross signalized intersections with different speed trajectories. The different speed trajectories are evaluated by calculating fuel usage, emissions, and travel time of vehicles through speed planning sections.

Fuel consumption in the uniform passing scenario is
(13)$$F_{b}=\exp\left(\sum_{i=0}^{3}k_{i,0}^{\left(e\right)}{v_{0}}^{i}\right)\times T_{b}$$
where $T_{b}$ is the time to pass the speed planning section at a constant speed with the original speed. The calculation formula is as follows
(14)$$T_{b}=\frac{L}{v_{0}}$$

Based on the MOE regression coefficients for NO_x_ and HC emissions fitted in [27,28], NO_x_ emissions $E_{{\mathrm{NO}}_{x},b}$ and HC emissions $E_{\mathrm{HC},b}$ can be calculated for the uniform speed through the intersection.

The acceleration function can be obtained by deriving the speed trajectory function, and the fuel consumption when accelerating or decelerating through the intersection can be obtained by substituting the speed function and acceleration function into the VT-Micro model. The formula is as follows
(15)$$\begin{array}{l} F\left(v_{c}\right)=\int_{0}^{\frac{\pi }{2\alpha }}\left(e^{\sum_{i=0}^{3}\sum_{j=0}^{3}\left(k_{i,j}^{e}\times {\left(v_{c}-(v_{c}-v_{0})\cos(\alpha t)\right)}^{i}\times {\left(\left(v_{c}-v_{0})\alpha \sin(\alpha t\right)\right)}^{j}\right)}\right)dt\\ +\int_{\frac{\pi }{2\alpha }}^{\frac{\pi }{2\alpha }+\frac{\pi }{2\beta }}\left(e^{\sum_{i=0}^{3}\sum_{j=0}^{3}\left(k_{i,j}^{e}\times {\left(v_{c}-(v_{c}-v_{0})\frac{\alpha }{\beta }\cos\beta (t-\frac{\pi }{2\alpha }+\frac{\pi }{2\beta })\right)}^{i}\times {\left(\left(v_{c}-v_{0})\alpha \sin\beta (t-\frac{\pi }{2\alpha }+\frac{\pi }{2\beta }\right)\right)}^{j}\right)}\right)dt+e^{\sum_{i=0}^{3}\left(k_{i,0}^{e}\times {\left(v_{c}+(v_{c}-v_{0})\frac{\alpha }{\beta }\right)}^{i}\right)}\times (\frac{L}{v_{c}}-\frac{\pi }{2\alpha }-\frac{\pi }{2\beta }) \end{array}$$

Based on the MOE regression coefficients for NO_x_ and HC emissions fitted in [27,28], NO_x_ emissions $E_{{\mathrm{NO}}_{x},b}$ and HC emissions $E_{\mathrm{HC},b}$ when accelerating or decelerating through the intersection can be obtained.

The time for accelerating or decelerating to pass the speed control section is
(16)$$T\left(v_{c}\right)=\frac{L}{v_{c}}$$

### 4.5. Target Speed Optimization Based on Genetic Algorithm

When the feasible speed set of the 1st to the kth intersection is not empty, there are multiple feasible target speeds *v*_c_, this paper uses a genetic algorithm to optimize the target speed. A genetic algorithm is a stochastic, probabilistic, intelligent search algorithm that simulates genetics in biological evolution. The algorithm deals with the encoding of variables in the solution set space and is not restricted by derivation or continuity. The genetic algorithm starts with the string set and covers a large area, which is conducive to global optimization, adopts an uncertain search direction, and measures the superiority of the solution by fitness calculation.

In this paper, the target speed is optimized to minimize emissions and fuel usage and maximize traffic efficiency (i.e., minimizing travel time). To better reflect the optimization effect of each optimization objective, the uniform speed passing scenario is defined as the base scenario. The objective function is established as
(17)$$J=w_{1}\times \frac{F\left(v_{c}\right)}{F_{b}}+w_{2}\times \frac{E_{NO_{X}}\left(v_{c}\right)}{E_{NO_{X},b}}+w_{3}\times \frac{E_{HC}\left(v_{c}\right)}{E_{HC,b}}+w_{4}\times \frac{T\left(v_{c}\right)}{T_{b}}$$
where *w*_1_, *w*_2_, *w*_3_, and *w*_4_ are weight coefficients, and their values are 0.3, 0.2, 0.2, and 0.3, respectively.

The target speed is selected in the feasible speed set, so the constraint is
(18)$$v_{c}\in V_{1}\cap V_{2}\cap \cdots \cap V_{k}$$

The fitness function is generally obtained by transforming the objective function, and the fitness also needs to be scaled so that the fitness value fits into the range of the selection function and to avoid the problem that individuals with high fitness values reproduce too quickly when the difference in fitness values is large and search is slow when the difference in fitness is small. The fitness is calculated, and the fitness is scaled by the method of Rank. The Rank method scales fitness according to the rank of each individual rather than the fitness value, with the most adapted individual (i.e., the one with the smallest objective function value) having a rank of 1, the next most adapted individual having a rank of 2, and so on, with the fitness scaling value of individuals with rank n being proportionate to $1/\sqrt{n}$. The sum of the fitness scaling values is equal to the number of parents in the next generation.

The selection operation uses a stochastic consistent method, sets the parents on a line, and adjusts the length corresponding to each parent individual according to the fitness scaling value. The algorithm moves along the line with the same size of steps, selecting a parent from the corresponding part for each step.

The crossover operation uses stochastic crossover to create a random binary vector that selects the first paternal gene, when the vector value is 1, and selects other paternal gene, when the vector value is 0, and combines the selected genes to make the offspring. The stochastic crossover schematic is shown in Figure 8.

The mutation operation places the current population of individuals in different rows and mutates each element with a probability of pm. The variation is introduced to give the genetic algorithm local stochastic search capability and to maintain the diversity of the population to prevent immature convergence.

Offspring individuals are produced in three ways: (1) individuals with the best fitness (i.e., the elite) will be retained directly into the next generation; (2) by the crossover of genes between two parent individuals; and (3) by mutation of a single parent individual.

To improve the solution efficiency while ensuring the solution accuracy, the parameters are as follows: The maximum number of iterations is 15, the population size is 15, the crossover probability is 0.8, and the variation probability is 0.2. Figure 9 shows the best individual fitness and the mean fitness for each generation, and it can be seen from the figure that the best individual fitness can converge within 15 generations with the parameters set in this study.

Figure 10 is the flowchart of the proposed strategy.

## 5. Experimental Analysis

### 5.1. Simulation Settings

This study uses MATLAB to connect VISSIM and build a simulation platform. MATLAB can obtain the simulation and evaluation information in the VISSIM road network through the COM interface, plan the corresponding speed trajectory using the eco-driving strategy and output it to the CAVs in the road network, then collate the acceleration and speed data obtained from the VISSIM simulation to evaluate the emission and fuel consumption. The architecture of the joint MATLAB and VISSIM simulation platform is shown in Figure 11.

The parameters are as follows: the study section is a one-lane straight-ahead section, and the section contains three signalized intersections. The signal cycle is 60 s, and the green light duration is 30 s. Based on the urban intersection capacity formula, the road capacity under the parameters set in this paper is about 800 veh∙h^−1^. Therefore, 500 veh∙h^−1^ representing medium traffic volume and 800 veh∙h^−1^ representing high traffic volume are set for simulation. The following model is using the Wiedemann 74 model. The simulation was run five times using various random seeds, and the average of those five runs was used to determine the result. Other parameters are shown in Table 1 and Figure 12. To better reflect the effect of the strategy, the result data is the data from the planning starting point to 150 m downstream of the 3rd intersection after 600 simulation seconds. The parameters used for evaluation include: the average speed of all vehicles in the study section (km/h), the total number of stops for all vehicles in the study section (times), the total delay due to stops for all vehicles in the study section (i.e., total stopping time, h), the total fuel consumption for all vehicles in the study section (L), NOx emissions (g), and HC emissions (g). These indicators can effectively assess traffic conditions, energy consumption, and emissions [11,18].

### 5.2. Eco-Driving Strategy Effectiveness Verification

Comparative experiments were conducted. The following driving strategies were used: the proposed strategy, strategy without speed optimization (Eco-driving strategy-so, which selects the speed with the smallest velocity variation in the feasible speed set as the target speed), strategy without speed optimization and without considering traffic flow effects (Eco-driving strategy-so-ctfe, which selects the speed with the smallest velocity variation in the feasible speed set as the target speed and sets queue length *l*_q_, queue dissipation time *t*_q_, and the length of the obstructed fleet *l*_h_ as 0), the eco-driving strategy with an optimization target only on fuel consumption and without considering the obstruction to following vehicles (Eco-driving strategy-fco-cofv, inspired by [21]), and a traffic flow without any driving strategy (manual driving vehicle, HDV) for comparison. Table 2 presents the comparison of the strategy in moderate traffic volume. From the table, the adoption of the eco-driving strategy leads to improvements. According to the comparison results between strategy-so-ctf and strategy-fco-cofv, it is evident that considering vehicle queues and optimizing speed trajectories based on fuel consumption significantly improves the effectiveness of the control strategies. The comparison between Eco-driving strategy and strategy-so and strategy-fco-cofv confirms the effectiveness of considering the obstruction to following vehicles. In the comparison between the strategy-so and the proposed strategy, the use of genetic algorithms for optimizing the target speed results in minimal average speed variation while improving other indicators significantly. This demonstrates the effectiveness of a strategy that considers the effects of queuing and obstruction to following vehicles, as well as speed optimization.

### 5.3. The Effect of Eco-Driving Strategy on Fuel Consumption Rate and Emission Rate

Figure 13 shows the comparison of vehicle fuel consumption rate, NO_x_ emission rate, and HC emission rate between the CAV flow (CAV penetration rate of 100%) and the HDV flow at medium traffic volume. The vehicle fuel consumption rate, NO_x_ emission rate, and HC emission rate are kept at a low level after adopting the eco-driving strategy, and the magnitude of periodic fluctuations is suppressed, so it contributes to the reduction of emissions and fuel consumption. This is because the eco-driving strategy enables vehicles to make full use of the green light time when crossing through multiple signal intersections, reducing unnecessary start-stop and acceleration/deceleration behaviors, thus reducing the proportion of high energy-emission working conditions and achieving energy saving and emission reduction.

### 5.4. The Effect of Eco-Driving Strategy on Traffic Performance and Fuel Consumption and Emissions

Figure 14 shows the comparison between the CAV flow and the HDV flow. As shown in the figure, CAV flow with an eco-driving strategy outperforms HDV flow in all indexes, and the optimization of the strategy is better than the high traffic volume when the traffic volume is moderate. When the traffic volume is 800 veh∙h^−1^, the road capacity is already saturated, so the strategy is not significantly effective in improving the average speed, but by optimizing the vehicle trajectory, it can still reduce the number of stops and stopping delays, thus reducing emissions and fuel usage. When the traffic volume is 500 veh∙h^−1^, the average speed increases by 9.49%, the number of stops is reduced by 78.17%, the stopping delay is reduced by 82.00%, and the fuel consumption and NO_x_ and HC emissions are reduced by 20.38%, 39.41%, and 46.61%, respectively.

### 5.5. The Effect of CAV Market Penetration Rate

The effect of penetration rate is investigated in this section. Figure 15 shows the traffic performance and fuel consumption emission benefits for different penetration rates at moderate traffic volume. As the penetration rate rises, the average speed improvement of mixed traffic flow decreases slowly and then increases, while the average speed improvement of HDVs continues to decrease; meanwhile, other traffic condition benefits and emissions and fuel usage benefits increase with the penetration rate rising. The reason is that when there is a low penetration rate, CAV vehicles accelerating through intersections are frequently impeded by HDVs, while decelerating through trajectories can be executed successfully, so the average speed decreases, and traffic performance benefits, fuel consumption, and emissions benefits in this situation are mainly attained by decelerating through. When there is a high penetration rate, the speed trajectory of CAVs is better executed and therefore achieves greater benefits. Except for the average speed, each index of HDVs can achieve benefits at various rates of penetration. The reason is that the trajectory optimization of CAVs on the road indirectly affects the speed trajectory of HDVs, which enables HDVs to optimize the speed trajectory and reduce stopping and unnecessary acceleration and deceleration through proper deceleration, which indicates that emission reduction and energy saving can be achieved as long as CAVs are introduced. When the penetration rate is 80%, the average speed of HDV decreases by 3.52%, but the number of stops decreases by 24.74%, the stopping delay decreases by 19.28%, and the fuel consumption, NO_x_, and HC emissions decrease by 1.92%, 6.65%, and 10.53%, respectively.

## 6. Conclusions

In this paper, we investigate the traffic flow situation, construct a traffic flow situation estimation model, optimize the feasible speed set that can enable vehicles to cross the intersection without stopping and without obstructing the following vehicles, generate different speed trajectories through trigonometric function curves that ensure passenger comfort, calculate the fuel consumption, emissions, and travel time of the speed trajectories, and establish a multi-objective optimization methodology to find the optimal speed trajectory. The results show that the proposed strategy improves vehicles’ fuel consumption, emissions, and traffic efficiency through multiple signalized intersections. The proposed strategy provides an effective and feasible technical solution for CAVs to cross through multiple signalized intersections.

This study assumed that all CAVs positions are accurate, and the network is instantaneous and reliable. Further research is needed to investigate the impacts of vehicle positioning errors, network delays, and potential effects of cyberattacks.

## Figures and Tables

**Figure 1 sensors-24-06356-f001:**
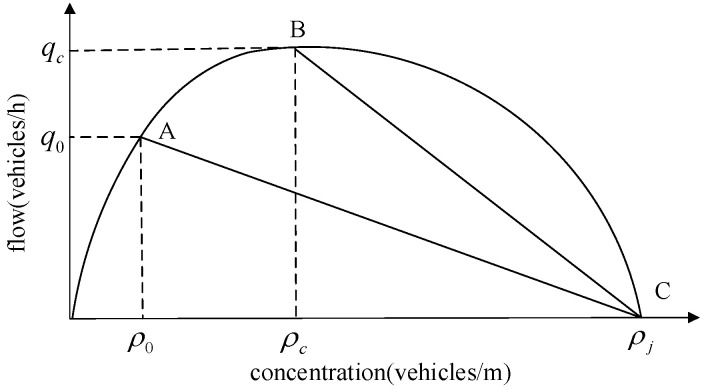
Basic relationship between traffic wave flow and concentration.

**Figure 2 sensors-24-06356-f002:**
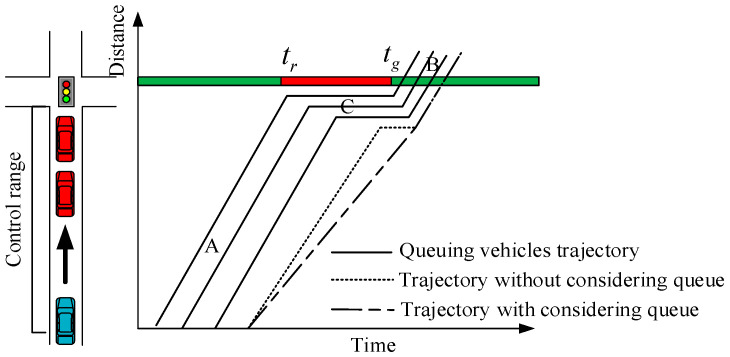
Schematic diagram of queuing at signalized intersections.

**Figure 3 sensors-24-06356-f003:**
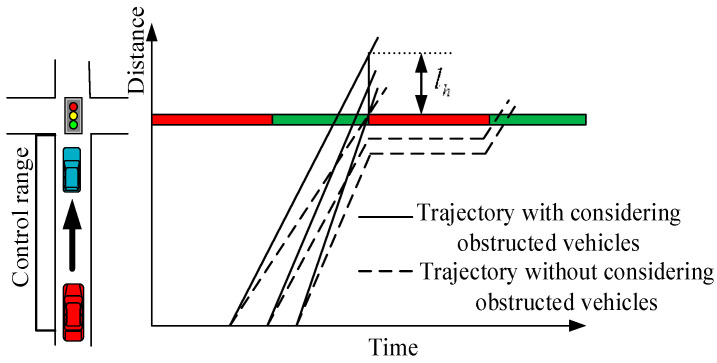
Schematic diagram of vehicles obstruction.

**Figure 4 sensors-24-06356-f004:**
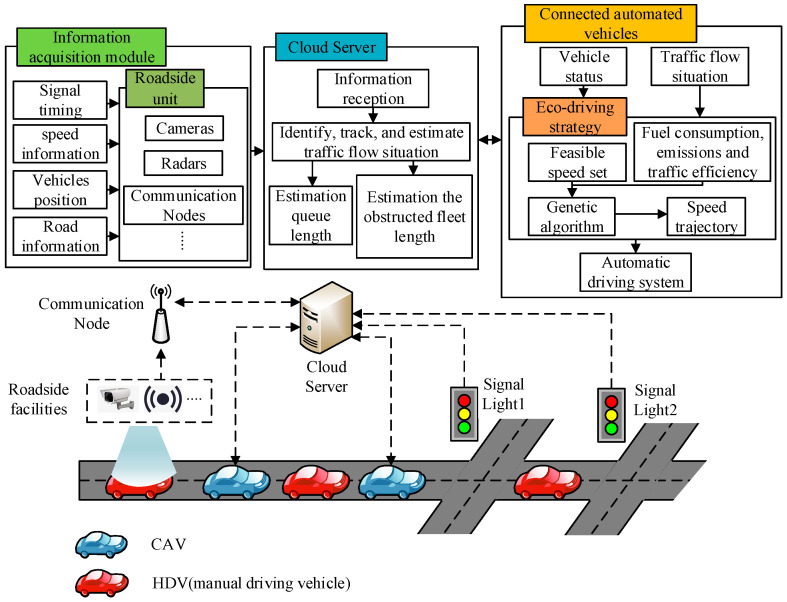
Eco-driving control system architecture.

**Figure 5 sensors-24-06356-f005:**
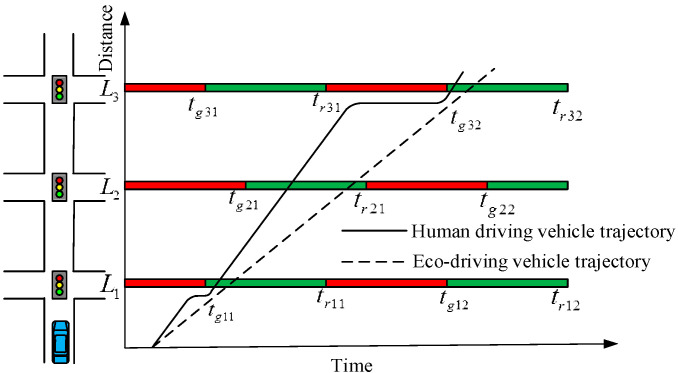
Schematic diagram of eco-driving vehicle trajectory.

**Figure 6 sensors-24-06356-f006:**
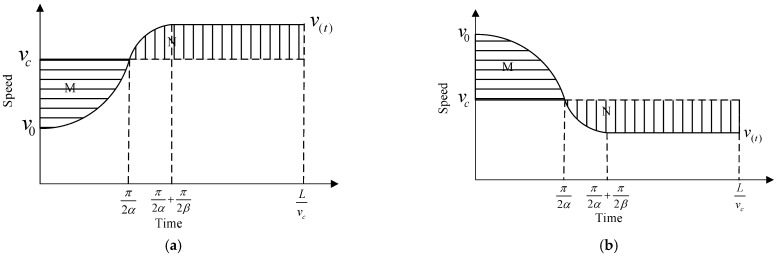
The speed trajectory of the trigonometric function curve. (**a**) Acceleration process; (**b**) Deceleration process.

**Figure 7 sensors-24-06356-f007:**
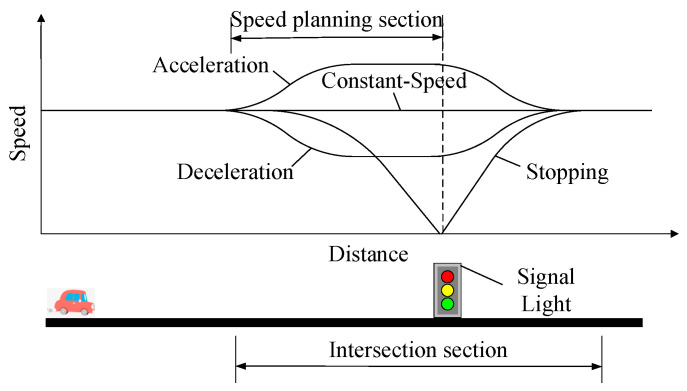
Schematic diagram of the connected eco-driving vehicle through the intersection.

**Figure 8 sensors-24-06356-f008:**
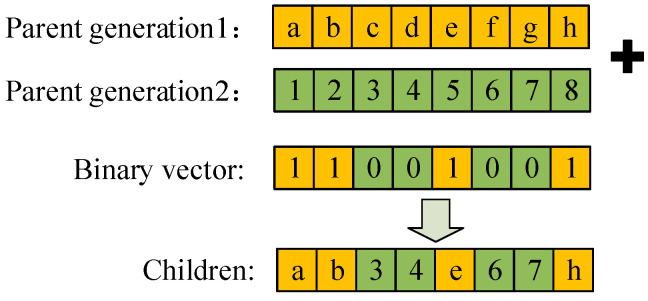
Stochastic crossover schematic.

**Figure 9 sensors-24-06356-f009:**
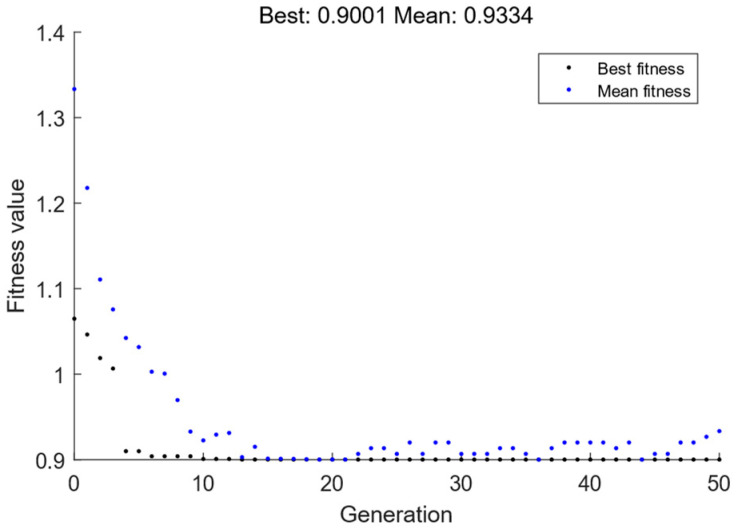
Best fitness value and mean fitness value of each generation.

**Figure 10 sensors-24-06356-f010:**
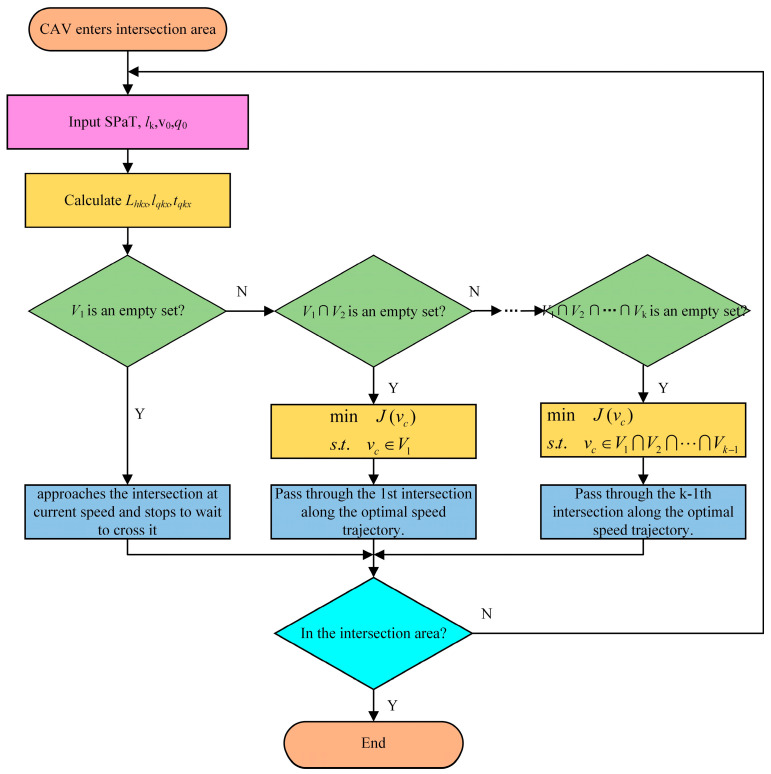
Flowchart of the proposed strategy.

**Figure 11 sensors-24-06356-f011:**
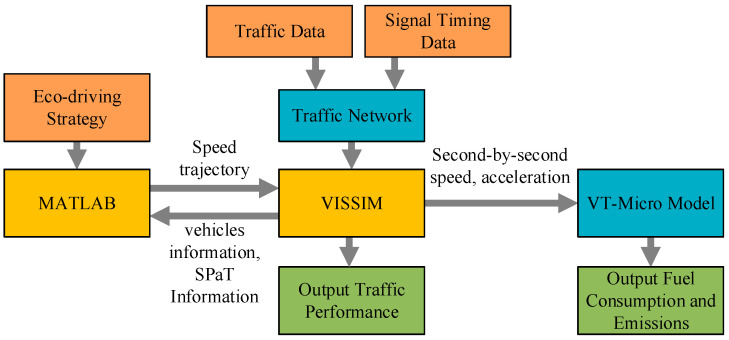
MATLAB and VISSIM joint simulation platform architecture.

**Figure 12 sensors-24-06356-f012:**
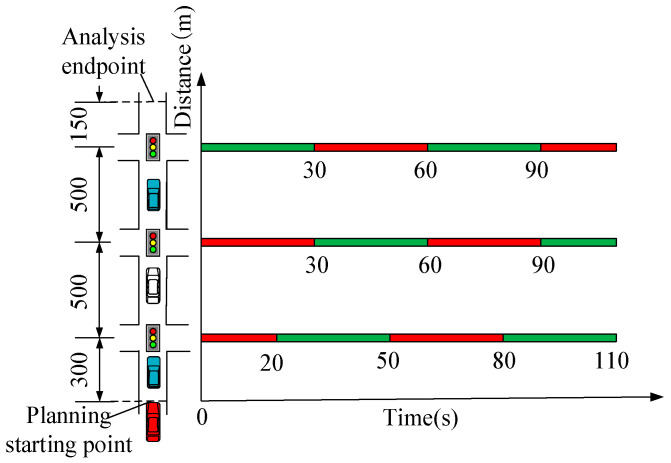
Simulation parameters diagram.

**Figure 13 sensors-24-06356-f013:**
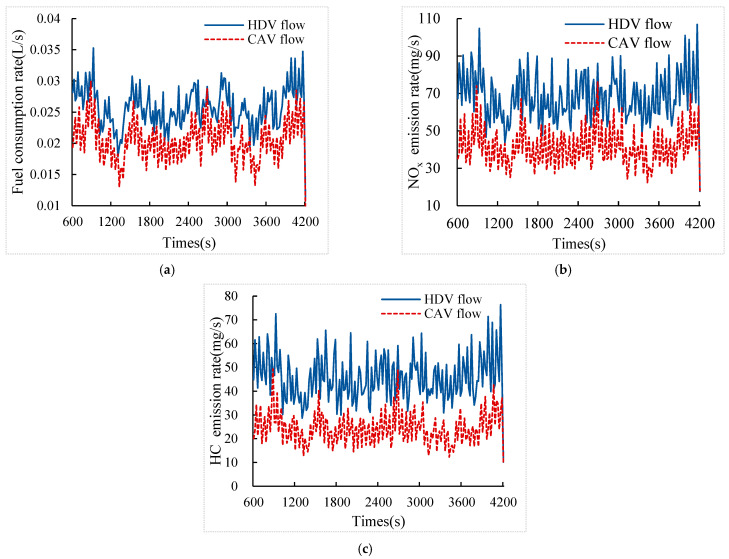
Fuel consumption rate and emission rate of CAV flow versus HDV flow at medium traffic volume. (**a**) Fuel consumption rate; (**b**) NO_x_ emission rate; (**c**) HC emission rate.

**Figure 14 sensors-24-06356-f014:**
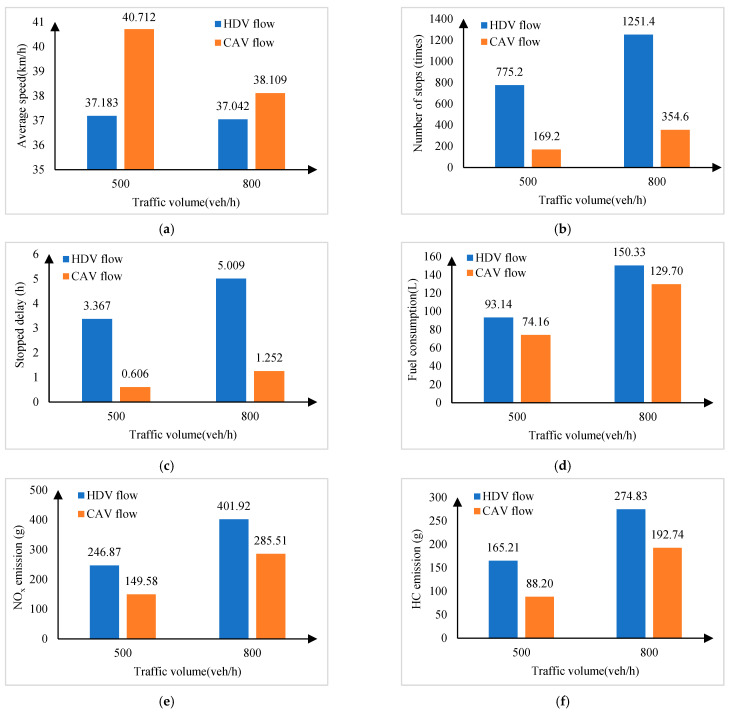
Comparison of traffic performance, fuel consumption, and emissions between CAV flow and manual driving traffic flow under different traffic volumes. (**a**) Average speed; (**b**) Number of stops; (**c**) Stopped delay; (**d**) Fuel consumption; (**e**) NO_x_ emission; (**f**) HC emission.

**Figure 15 sensors-24-06356-f015:**
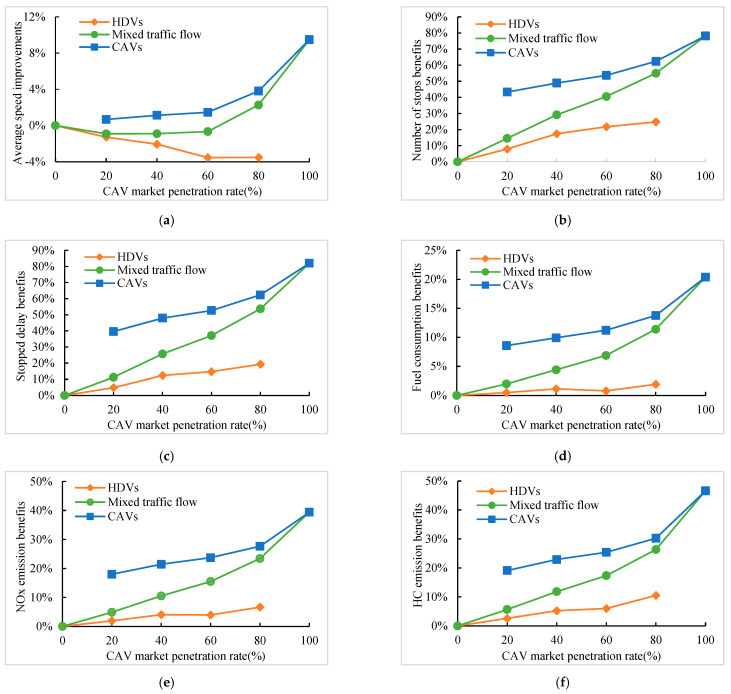
Benefits for different penetration rates. (**a**) Average speed improvement; (**b**) Number of stops benefits; (**c**) Stopped delay benefits; (**d**) Fuel consumption benefits; (**e**) NO_x_ emission benefits; (**f**) HC emission benefits.

**Table 1 sensors-24-06356-t001:** Simulation parameters.

Parameter	Value
Control section starting point location (m)	400
Location of the first intersection (m)	700
Location of the second intersection (m)	1200
Location of the third intersection (m)	1700
Location of analysis endpoint (m)	1850
T_g11_ (s)	20
T_r11_ (s)	50
T_g21_ (s)	30
T_r21_ (s)	60
T_g31_ (s)	0
T_r31_ (s)	30
Maximum speed limit of the road (km·h^−1^)	70
Minimum control speed (km·h^−1^)	30
Free flow speed (km·h^−1^)	50
Simulation duration (Simulation seconds)	4200

**Table 2 sensors-24-06356-t002:** Comparison of the eco-driving strategy effects.

	Average Speed /(km/h)	Number of Stops /(times)	Stopped Delay /(h)	Fuel Consumption /(L)	NO_x_ Emission /(g)	HC Emission /(g)
HDV flow	37.183	775.2	3.367	93.14	246.87	165.21
Eco-driving strategy -so-ctfe	38.323	314.4	1.529	82.80	194.98	127.33
Eco-driving strategy -fco-cofv	39.392	191.6	0.755	77.52	159.72	101.07
Eco-driving strategy -so	40.723	176.6	0.642	75.22	156.56	92.47
Eco-driving strategy	40.712	169.2	0.606	74.16	149.58	88.20

## Data Availability

Data are contained within the article.

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
