# Peer review of "An Eco-Driving Strategy at Multiple Fixed-Time Signalized Intersections Considering Traffic Flow Effects"

_sensors, 2024, doi:10.3390/s24196356_

Round 1

Reviewer 1 Report

Comments and Suggestions for Authors

Comment to

Manuscript An Eco-driving Strategy at Multiple Fixed-time Signalized Intersection
Considering Traffic Flow Effects
  by  Authors: Huinian Wang, Jingyao Wang, Jinghua Guo  submitted  to Sensors,

The topic of research is a very relevant and important problem The primary question addressed by this research is: How can an eco-driving strategy for connected and automated vehicles (CAVs) be developed and optimized to enhance traffic efficiency while reducing fuel consumption and emissions at multiple fixed-time signalized intersections, considering the dynamics of mixed traffic environments?  The research aims to formulate a comprehensive approach that integrates the interactions between connected and automated vehicles (CAVs) and general traffic flow, optimizes speed trajectories, and assesses the strategy's effectiveness under real-world traffic conditions.

 The paper introduces an innovative traffic flow situation estimation model that estimates queue lengths and the length of obstructed fleets at multiple signalized intersections. This model is critical for accurately predicting traffic conditions and optimizing vehicle trajectories, contributing significantly to the field of traffic management.    A novel method is presented to calculate feasible speed sets, enabling vehicles to pass through multiple intersections without stopping. This approach enhances traffic flow and riding comfort by employing a trigonometric profile for speed trajectory planning, marking an advancement in the eco-driving strategies for connected and automated vehicles (CAVs).    The research makes a significant contribution by employing genetic algorithms to optimize speed trajectories. This method balances multiple objectives, such as fuel consumption, emissions, and traffic efficiency, offering a holistic approach to eco-driving that integrates environmental and efficiency considerations.

The use of MATLAB and VISSIM for simulating and validating the proposed strategy adds practical relevance to the research. It demonstrates the strategy's effectiveness in realistic traffic scenarios, which is essential for potential real-world applications, thus bridging the gap between theoretical models and practical implementation.

The paper addresses a notable gap in the field by focusing on the interaction between CAVs and nonautomated vehicles in a mixed traffic environment. Previous research predominantly targets isolated aspects, such as vehicle-to-infrastructure communication or optimizing speed for fuel efficiency, without fully considering the complex dynamics of mixed traffic conditions.

   The strategy's validation through simulations that mimic real-world conditions bridges the gap between theoretical research and practical, implementable solutions for urban traffic management. This practical relevance is essential for the adoption and success of eco-driving strategies in real-world applications.

   While the study considers medium and high traffic volumes, it could benefit from including a wider range of traffic scenarios, such as low traffic volume or rush-hour congestion. This would provide a more comprehensive evaluation of the strategy's robustness and adaptability to different traffic conditions.

   Integrating real-world traffic data from various urban settings could enhance the accuracy and applicability of the simulation results. Using actual traffic patterns, vehicle behaviors, and signal timings would help validate the model under realistic conditions, improving its practical relevance.

   Extending the simulation duration to evaluate the strategy's performance over longer periods (e.g., several days or weeks) would provide insights into its long-term benefits and potential drawbacks, ensuring that the strategy remains effective over time.

   Including various types of intersections in the study could assess how the strategy performs in different settings. This would determine if specific configurations require additional modifications, enhancing the strategy's versatility.

   Conducting a sensitivity analysis on key parameters, such as signal timing, vehicle speed limits, and CAV penetration rates, would help identify critical factors that influence the strategy's effectiveness. This analysis would provide valuable guidance for finetuning the model.

How can an eco-driving strategy for CAVs be developed and optimized to improve traffic efficiency and reduce fuel consumption and emissions at multiple fixed-time signalized intersections, considering the effects of traffic flow in a mixed traffic environment?

 Development and validation of a model to estimate intersection queue lengths and obstructed fleet lengths. Accurate estimations achieved, supporting the feasibility of planning speed trajectories.

Calculation of speed sets allowing vehicles to pass multiple intersections without stopping.

Simulations demonstrated smooth navigation through intersections, reducing stops and delays.

Optimization of speed trajectories based on fuel consumption, emissions, and traffic efficiency using genetic algorithms.

    Optimized trajectories led to significant improvements in traffic flow and environmental metrics, confirming the approach's effectiveness.

Running simulations with MATLAB and VISSIM to test the strategy under various traffic conditions.

 Validation through simulations confirmed improvements in average speeds, reduced stops, and lower emissions and fuel consumption.

These experiments provide robust evidence supporting the paper's claims about the effectiveness and practicality of the proposed eco-driving strategy for CAVs in mixed traffic environments.

The references included in the paper "An Eco-driving Strategy at Multiple Fixed-time Signalized Intersections Considering Traffic Flow Effects" appear to be appropriate and relevant. They cover a broad spectrum of previous research related to eco-driving strategies, CAVs, signalized intersections, and optimization methods. Here are some points of comparison and assessment: The references encompass critical areas such as eco-driving algorithms, connected and automated vehicle (CAV) technologies, optimization techniques, and environmental impact assessments.    Key studies on speed planning for signalized intersections, cooperative control of CAVs, and multi-objective optimization are cited, ensuring a comprehensive literature review.

The paper cites recent research articles, which ensures that the strategy proposed is built upon the latest advancements and findings in the field.

References include studies from 2021 and 2022, demonstrating the paper's alignment with contemporary research.

The references encompass critical areas such as eco-driving algorithms, connected and automated vehicle (CAV) technologies, optimization techniques, and environmental impact assessments. Key studies on speed planning for signalized intersections, cooperative control of CAVs, and multi-objective optimization are cited, ensuring a comprehensive literature review. The cited works include specific contributions relevant to the paper’s focus, such as queue length estimation, fuel consumption minimization, and emission reduction. These are crucial for developing effective eco-driving strategies for CAVs. Overall, the references in the paper are appropriate and provide a solid foundation for the proposed eco-driving strategy.

Questions

Please, explain how you measured the fuel consumption, NOx and HC emissions in experiment?

Is it possible to compare results with other research in this field?

The article is written in an understandable language, not overloaded with unnecessary terminology. The conclusions of the authors are well founded. The article is of practical interest and is recommended for publication after minor revision.

Reviewer 2 Report

Comments and Suggestions for Authors

The Title: An Eco-driving Strategy at Multiple Fixed-time Signalized Intersection Considering Traffic Flow Effects.

In this paper, an eco-driving strategy for connected and automated vehicles (CAVs) considering the effects of traffic flow is proposed for the mixed traffic environment. The work is good, but there are some major points need to be considered which are as follows:

1- First of all, the contributions of the work are somehow weak and need to be highlighted more in the Abstract.

2- In the introduction section, more introductions about this field using recent works need to be added. Moreover, it is also necessary to emphasize the need for an eco-driving and more efficient strategy to enable practical solutions for CAVs in real-world scenarios.

3- Presentation of related work is weak and needs to be improved. It is also recommended to show the weak points in these works to highlight the purpose of the current work.

4- Any figures, tables, information should be cited with reliable source, unless they belong to the authors. Please check such issues for the entire manuscript.

5- A flowchart is required to show the workflow of the proposed algorithm while providing details of each step.

6- There are some typos that need to be corrected and further organization of the structure is recommended. For example, “NOx emissions” is written in different forms in the text body of the manuscript.

Moreover, it is recommended to write the full term of the abbreviation in the first appearance for them for example, “HDV flow”. Please correct this issue for the whole manuscript.

7- Why are three signalized intersections used in the study section? please clarify the reason?

8- Can the authors provide a simple network among the signalized intersections?

9- Please identify the parameters used for the purpose of evaluation along with their units (if any) and provide a reliable source for them.

10- The comparison in Table 2 was made based on a self-comparison of the current work by changing some parameters.

11- It is recommended to compare the proposed work with other recent works to demonstrate the extent to which its performance has improved compared to other works and the new features that have been added to this field.

12- The conclusion section should be enhanced and future trends need to be added.

Comments on the Quality of English Language

Minor editing of English language required

Round 2

Reviewer 1 Report

Comments and Suggestions for Authors

Thank you so much for improving the manuscript

Reviewer 2 Report

Comments and Suggestions for Authors

Comments have been addressed as requested and no further comments are needed.

Comments on the Quality of English Language

Minor editing of English language required